# Transmembrane Protein 175, a Lysosomal Ion Channel Related to Parkinson’s Disease

**DOI:** 10.3390/biom13050802

**Published:** 2023-05-09

**Authors:** Tuoxian Tang, Boshuo Jian, Zhenjiang Liu

**Affiliations:** 1National Engineering Laboratory for AIDS Vaccine, School of Life Sciences, Jilin University, Changchun 130012, China; 2Department of Biology, University of Pennsylvania, Philadelphia, PA 19104, USA

**Keywords:** TMEM175, potassium channel, proton channel, lysosome, Parkinson’s disease

## Abstract

Lysosomes are membrane-bound organelles with an acidic lumen and are traditionally characterized as a recycling center in cells. Lysosomal ion channels are integral membrane proteins that form pores in lysosomal membranes and allow the influx and efflux of essential ions. Transmembrane protein 175 (TMEM175) is a unique lysosomal potassium channel that shares little sequence similarity with other potassium channels. It is found in bacteria, archaea, and animals. The prokaryotic TMEM175 consists of one six-transmembrane domain that adopts a tetrameric architecture, while the mammalian TMEM175 is comprised of two six-transmembrane domains that function as a dimer in lysosomal membranes. Previous studies have demonstrated that the lysosomal K^+^ conductance mediated by TMEM175 is critical for setting membrane potential, maintaining pH stability, and regulating lysosome–autophagosome fusion. AKT and B-cell lymphoma 2 regulate TMEM175’s channel activity through direct binding. Two recent studies reported that the human TMEM175 is also a proton-selective channel under normal lysosomal pH (4.5–5.5) as the K^+^ permeation dramatically decreased at low pH while the H^+^ current through TMEM175 greatly increased. Genome-wide association studies and functional studies in mouse models have established that TMEM175 is implicated in the pathogenesis of Parkinson’s disease, which sparks more research interests in this lysosomal channel.

## 1. Introduction

Lysosomes (and equivalent vacuoles in plant and yeast cells) are membrane-enclosed cytoplasmic organelles that contribute to the breakdown and recycling of biological macromolecules in nearly all eukaryotic cells. The cargo vesicles originating from endocytic, phagocytic, and autophagic pathways are sent to lysosomes for degradation, and the resulting catabolic products are exported to the cytoplasm for subsequent reutilization [1,2,3]. Growing evidence shows that lysosomes are more than a terminal recycling center in cells. They serve as a signaling hub that integrates signals from multiple metabolic pathways and cellular structures, which further associates this acidic organelle with a plethora of cellular functions [4,5]. More than 60 hydrolases are found in their acidic lumen, which is a hallmark of lysosomes. The luminal low pH environment is limited by a single-lipid bilayer membrane embedded with a wide variety of important proteins including transporters, pumps, and ion channels, which control the influx/efflux of metabolites and ions [6]. For example, the vacuolar H^+^-ATPase (V-ATPase) is an ATP-dependent proton pump that transports protons across the lysosomal membrane to establish and maintain the acidic pH of the lysosomal lumen [7].

Ion channels are integral pore-forming membrane proteins that facilitate the passive movement of ions across the membrane. They play a critical role in nearly all fundamental physiological processes, including muscle contraction, synaptic transmission, and action potential propagation. Therefore, it is not surprising that mutations in lysosomal channels give rise to a wide range of diseases, such as renal disorders, heart diseases, and neurodegenerative diseases [8,9]. Several types of ion channels have been identified on lysosomal membranes [10,11]. The still-expanding list includes the non-selective cation channels, transient receptor potential mucolipins (TRPMLs) [12,13,14], volume-regulated Cl^−^/anion channel, leucine-rich repeat-containing family 8 (LRRC8) [15], endolysosome-localized chloride channel, H^+^/Cl^−^ exchanger 7 (CLC-7) [16], Na^+^/Ca^2+^-permeable P2 × 4 purinoceptor [17], Na^+^-selective two-pore channels (TPCs) [18,19,20], K^+^-selective channels such as the large conductance Ca^2+^-activated K^+^ channel (big potassium, BK) [21,22], and transmembrane protein 175 (TMEM175) [23].

Potassium ions are the most abundant cations inside the cytosol and potassium channels are widely distributed in numerous cell types. Based on their gating mechanisms and structural characteristics, the K^+^ channels are classified into several categories including Ca^2+^-activated K^+^ channels (K_Ca_), voltage-gated K^+^ channels (K_v_), inwardly rectifying K^+^ channels (K_ir_), tandem pore domain K^+^ channels (K_2P_), and Na^+^-activated K^+^ channels [24,25,26]. TWIK2, BK, and TMEM175 are the three lysosomal K^+^ channels that have been discovered thus far. The K2P family member TWIK2 (tandem of pore domains in a weakly inward rectifying K^+^ channel 2) generates weak background K^+^ currents in lysosomes [27]. BK channels exhibit a high K^+^ selectivity and a large single-channel conductance, which confer a significant physiological mechanism to modulate lysosomal membrane potential and intracellular Ca^2+^ homeostasis [21,22]. TMEM175 has been identified as a K^+^-selective channel on endosomes and lysosomes, which remarkably contributes to the total K^+^ conductance in lysosomes [23,28]. Crystal structures of prokaryotic TMEM175 from *Chamaesiphon minutus* [29] and *Marivirga tractuosa* [30] and cryo-EM structures of human TMEM175 [31,32] have suggested that TMEM175 shows a different structure and K^+^ selectivity mechanism from canonical K^+^ channels. As a major functional K^+^ channel of lysosomes, TMEM175 plays a vital role in setting lysosomal membrane potential and maintaining lysosomal pH stability [23].

TMEM175 is a growth-factor-activated and AKT (protein kinase B)-gated lysosomal ion channel. It forms a complex with AKT and conformational changes in AKT are sufficient to activate TMEM175 [28]. A recent study reported that the apoptosis regulator Bcl-2 (B-cell lymphoma 2) binds to TMEM175 and suppresses its ion channel activity, which reveals the involvement of TMEM175 in apoptotic pathways [33]. Further data point to TMEM175 being more than just a K^+^ channel. It exhibits proton selectivity and conductivity, generating a constitutive proton current and acting as a lysosomal H^+^-leak channel [32,34].

TMEM175 has been recognized as a genetic risk factor for Parkinson’s disease (PD) by genome-wide association studies [35,36]. PD is the second most prevalent neurodegenerative disorder in the elderly, and it is clinically characterized by motor disabilities of rigidity, tremor, and bradykinesia as well as many non-motor symptoms including cognitive decline, sleep disorder, depression, olfactory loss, and autonomic dysfunction. The primary neuropathology of PD features the loss of dopaminergic neurons in the *substantia nigra pars compacta* and α-synuclein accumulation [37,38,39]. TMEM175 deficiency causes accumulation of α-synuclein in neurons and loss of dopaminergic neurons, which are prominent pathological characteristics of PD [28,34,40].

## 2. Structures of TMEM175

TMEM175 channels are found in archaea, bacteria, and animals, but are not present in fungi and plants. They differ from canonical K^+^ channels in several aspects. First, they share little amino acid sequence homology with other known potassium channels, rendering them evolutionarily distinct. Second, most canonical K^+^ channels feature a conserved P-loop selectivity filter composed of TVGYG-like signature sequences [41,42], but this conserved motif is missing in TMEM175 channels, which may indicate that they show a great difference in ion permeation properties. Third, unlike the canonical K^+^ channels that assemble the C-terminal helix as a pore-lining helix, TMEM175 channels assemble their subunits in an inverted way, with the N-terminal helix serving as a pore-lining helix [43,44]. Lastly, TMEM175 channels and canonical K^+^ channels show distinct pharmaceutical properties.

Prokaryotic TMEM175 proteins display a low sequence identity (Figure 1A). The sequence identity between CmTMEM175 and MtTMEM175 is lower than 20%. The most conserved region in prokaryotic TMEM175 proteins is the transmembrane 1 (TM1) helix that lines the channel pore. In contrast, mammalian TMEM175 channels show a much higher sequence identity. The human TMEM175 and the gorilla TMEM175 almost have an identical protein sequence. Additionally, the sequence identity between human TMEM175 and mouse TMEM175 is as high as 81% (Figure 1B). However, the level of conservation between prokaryotic and eukaryotic TMEM175 homologs is very low. The sequence identity between CmTMEM175 and hMEM175 repeat I or repeat II is 17.3% and 20.8%, respectively (Figure 1A).

The structures of three different TMEM175 proteins have been resolved so far: two prokaryotic TMEM175 homologs from *Chamaesiphon minutus* (CmTMEM175) [29] and *Marivirga tractuosa* (MtTMEM175) [30], and the human homolog (hTMEM175) [31,32]. Even though prokaryotic and eukaryotic TMEM175 proteins do not share many similarities in their amino acid sequences, these three TMEM175 variants display a similar structural topology. The prokaryotic TMEM175 proteins exhibit a homotetrameric architecture, with each protomer containing one repeat of six transmembrane (6-TM) helices (Figure 2A,B). The first TM helix of each of the four protomers forms a central ion permeation channel, which is in line with the finding that the TM1 helix exhibits the highest level of conservation [29,30]. The human TMEM175, on the other hand, is a homodimer and each subunit contains two repeats of 6-TM helices (repeat I and repeat II) (Figure 2C,D) [31,32]. The structures of repeat I and repeat II highly resemble the monomeric structures of prokaryotic TMEM175 channels. The first transmembrane helix of repeat I (TM1) and repeat II (TM7) together form the ion-permeation pore. The ion-conduction pathway is situated in the middle of the channel along the pseudo-four-fold axis of the human TMEM175, which adopts a pseudo-four-fold symmetric architecture [31,32].

As TMEM175 channels lack the canonical P-loop selectivity filter, it raises the question of how they achieve K^+^ permeation and selectivity. In CmTMEM175, four TM1 helices form a bundle crossing that contains three layers of highly conserved hydrophobic residues. It was proposed that the conserved Ile23, Leu27, and Leu30 residues from each pore-lining TM1 helix create an ion permeation pathway, with the Ile23 residue playing a key role in defining K^+^ permeability and selectivity [29]. In line with this proposal, two independent studies have demonstrated that the conserved Ile46 from TM1 and the conserved Ile271 from TM7 constitute a narrow hydrophobic isoleucine constriction that acts as a physical gate and ion selectivity filter in hTMEM175 [31,32,45].

However, the structural and functional investigation of MtTMEM175 offers a divergent hypothesis for how TMEM175 channels selectively permeate potassium ions, despite having somewhat similar overall structures. Instead of hydrophobic residues, a highly conserved layer of threonine (Thr38) was discovered to be responsible for the K^+^ selectivity of MtTMEM175. The bulky hydrophobic residue Leu35 at the homologous position of Ile23 in CmTMEM175 or Ile46/Ile271 in hTMEM175 serves as a physical gate rather than a selectivity filter for ion permeation [30]. Furthermore, a layer of threonine residues (Thr49 and Thr274) and a serine residue (Ser45) also contribute to the K^+^ selectivity of hTMEM175. Selective permeation of ions through ion channels largely depends on the interplay between channel–ion interactions and water–ion interactions. The behavior of water within an ion channel pore is also very important to its conductive status. A “hydrophobic gating” mechanism may exist in TMEM175 channels, through which the hydrophobic pore spontaneously dehydrates and thereby functionally closes the channel [46,47].

The human TMEM175 shows decreased K^+^ permeation and increased H^+^ conductance at an acidic pH. It has been shown via molecular dynamics simulation and structure-guided mutagenesis that protons and potassium ions may share a permeation pathway. Besides the conserved amino acid residues that determine the K^+^ selectivity of TMEM175 channels, the N-terminal region of TM1 has a conserved RxxxFSD motif (Figure 1A,B). These residues are involved in inter- and intra-subunit interaction networks, which play a crucial role in maintaining the quaternary structure of TMEM175 channels [29,31]. Mutations in this signature motif abolished the channel activity [23].

## 3. The Ion Selectivity of TMEM175

### 3.1. The Human TMEM175 Is Characterized as a Lysosomal K^+^-Selective Channel

TMEM175 was first identified as a multi-spanning transmembrane protein of unknown functions (Domain-of-Unknown-Function DUF1211) through comparative proteomic analysis of integral and associated lysosomal membrane proteins [48]. The subcellular localization of TMEM175 was confirmed via transient expression of YFP-fusion protein in HeLa cells and compared with the lysosomal/late endosomal marker LAMP1 [49]. A whole-organelle patch-clamp-based candidate gene screening was utilized to identify the lysosomal K^+^ channel that is responsible for the K^+^ conductance in endosomes and lysosomes. Lysosomal currents were recorded from candidate genes-transfected HEK293T (human embryonic kidney 293T) cells. Of the 16 screened candidates, only TMEM175 showed a large increase in potassium current. TMEM175 contributes to the major lysosomal K^+^ conductance. The knock-down of mouse TMEM175 (mTMEM175) in RAW264.7 cells (a murine macrophage cell line commonly used in lysosomal studies) and human TMEM175 (hTMEM175) in HEK293T cells with short hairpin interfering RNAs significantly decrease the native K^+^ conductance. Furthermore, the knock-out of mTMEM175 in RAW264.7 cells with the CRISPR/Cas9 technique abolished the K^+^ conductance, which suggests that TMEM175 confers the lysosomal K^+^ permeability [23]. Ion selectivity is a fundamental property of ion channels and crucial to their physiological functions. hTMEM175 is permeable to Rb^+^, Cs^+^, and K^+^. It shows a better permeability for Cs^+^ than K^+^, which is consistent with the ion permeability of the lysosomal membrane [50]. It is minimally permeable to NMDG^+^ (N-methyl-D-glucamine, a large organic cation), Na^+^, and Ca^2+^ (Table 1).

TMEM175 homologs are also found in bacteria and archaea. The cellular localization of prokaryotic TMEM175 channels is the plasm membrane as prokaryotes do not have intracellular organelles. The ion selectivity of several prokaryotic TMEM175s has been analyzed through overexpressing bacterial TMEM175 proteins in HEK293T cells and using whole-cell patch clamp recording in electrophysiological characterization. *Chryseobacterium* sp. TMEM175 (CbTMEM175) and *Streptomyces collinus* TMEM175 (ScTMEM175) are more permeable to Na^+^ and less permeable to Cs^+^ than hTMEM175 [23]. Similar to CbTMEM175 (P_K_/P_Na_ ≈ 2.4 ± 0.5, Table 1) and ScTMEM175 (P_K_/P_Na_ ≈ 4.4 ± 1.4, Table 1), the *Marivirga tractuosa* TMEM175 (MtTMEM175) is a weakly K^+^ selective channel (P_K_/P_Na_ ≈ 4.3, Table 1) [30]. Using 86Rb assay, another prokaryotic TMEM175 channel, *Chamaesiphon minutus* TMEM175 (CmTMEM175), was found to be selective for K^+^, Rb^+^, and Cs^+^. This channel is not permeable to NMDG^+^ or Na^+^, which is consistent with the ion selectivity of hTMEM175 (Table 1) [29]. The physiological functions of prokaryotic TMEM175 channels remain unknown but they are supposed to be responsible for the influx/efflux of specific ions depending on their ion selectivity. Overall, hTMEM175 shows higher selectivity for K^+^ and lower Na^+^ permeability than prokaryotic TMEM175s.

**Table 1 biomolecules-13-00802-t001:** The ion selectivity of TMEM175.

TMEM175s	Ion Selectivity	Permeable Ions	References
hTMEM175	P_K_/P_Na_ = 36.0 ± 4.4 P_K_/P_Ca_ = 141.6 ± 27.7 P_K_/P_Cs_ = 0.51 ± 0.03	K^+^, Cs^+^, Rb^+^	[23]
hTMEM175	P_K_/P_Na_ ≈ 9.0	K^+^, Cs^+^	[31]
CbTMEM175	P_K_/P_Na_ = 2.4 ± 0.5 P_K_/P_Cs_ = 2.1 ± 0.7	K^+^, Cs^+^, Na^+^	[23]
ScTMEM175	P_K_/P_Na_ = 4.4 ± 1.4 P_K_/P_Cs_ = 2.3 ± 0.3	K^+^, Cs^+^, Na^+^	[23]
CmTMEM175	P_K_ > P_Na_	K^+^, Cs^+^, Rb^+^	[29]
MtTMEM175	P_K_/P_Na_ ≈ 4.3	K^+^, Cs^+^, Rb^+^, Na^+^, Li^+^	[30]
hTMEM175	P_H_/P_K_ ≈ 3.3 × 10^4^	H^+^, Cs^+^, K^+^	[32]
hTMEM175	P_H_/P_K_ ≈ 4.8 × 10^4^	H^+^, K^+^	[34]

Abbreviations: hTMEM175: human TMEM175; CbTMEM175, *Chryseobacterium* sp. TMEM175; ScTMEM175, *Streptomyces collinus* TMEM175; CmTMEM175, *Chamaesiphon minutus* TMEM175; MtTMEM175, *Marivirga tractuosa* TMEM175. P_K_, P_Na_, P_Ca_, P_Cs_, and P_H_ are the permeabilities to K^+^, Na^+^, Ca^2+^, Cs^+^, and H^+^, respectively.

### 3.2. The Human TMEM175 Emerges as a Lysosomal H^+^-Leak Channel

The catabolic function of lysosomes is accomplished by more than 60 digestive enzymes including proteases, peptidases, lipases, nucleases, glycosidases, and sulfatases that break down biological macromolecules into small building-block molecules. These hydrolytic enzymes require an acidic pH of ~4.7 to carry out their catalytic activities. The acidity of the lysosomal lumen is dynamically and meticulously regulated in order to achieve pH homeostasis. Disturbance of lysosomal pH can cause lysosomal dysfunctions which are associated with multiple diseases such as lysosomal storage disorders, Parkinson’s disease, and Alzheimer’s disease [51,52,53]. Many ion channels, pumps, and transporters may contribute to the pH homeostasis of lysosomes in different ways.

It is well established that the vacuolar H^+^-ATPase, also known as the V-ATPase, is an ATP-dependent proton pump that mediates the active transport of protons from the cytoplasm to the lysosomal lumen and propels lysosome acidification [54,55,56]. A transmembrane electrical potential difference forms between the lysosome and cytosol when a significant number of protons have been pumped into the lysosomal lumen, which precludes additional proton pumping and consequently restricts the acidification of lysosomes. To circumvent this self-limiting behavior, proton influx must be accompanied by the movement of a counterion to dissipate the transmembrane voltage. Either the influx of cytosolic anions into the lysosomal lumen, the efflux of luminal cations into the cytosol, or the concurrent activity of both pathways can divert the transmembrane voltage generated by V-ATPases and facilitate lysosomal acidification [54,57]. The Cl^−^/H^+^ antiporter ClC-7 transports chloride ions into the lysosomal lumen and moves protons out of lysosomes with a fixed stoichiometry of 2Cl^−^:1H^+^, which has an important role in lysosomal pH regulation [16,57,58]. Na^+^ and K^+^ channels may also participate in counterion pathways and give rise to the effective acidification of lysosomes. For instance, an endolysosomal ATP-sensitive Na^+^ channel formed by two-pore channels (TPCs) can regulate lysosomal pH stability [19,59].

Early studies on the acidification of endosomes and lysosomes have revealed a proton leak current in these cellular compartments [60]. Additionally, a proton leak current was also detected in organelles of the regulated secretory pathway [61], which indicates that an unidentified proton-leak channel is present on endolysosomal membranes. Proton leak is proposed to be an important determinant of lysosomal pH, but the molecular identity of this lysosomal proton leak channel remains a mystery [34,57].

Two recent studies have demonstrated that hTMEM175 is a proton-selective and proton-activated channel that mediates the proton leak current of lysosomes [32,34]. A “proton-leak” like current was observed in hTMEM175-overexpressing HEK293T cells, but not in hTMEM175-KO (knock-out) cells, which suggests that hTMEM175 is responsible for the proton conductance. hTMEM175 is highly selective and permeable to protons when the luminal side is exposed to a very acidic pH. In contrast, it displays decreased K^+^ permeation at low pH [32]. It is estimated that the permeability of hTMEM175 for H^+^ relative to K^+^ is at a ratio of about 0.5 × 10^5^-fold (Table 1).

### 3.3. Pharmacological Properties of TMEM175

Reflecting their divergent pore structure and selectivity profile, TMEM175 channels show highly distinct pharmacological properties compared to canonical K^+^ channels. First, unlike many K^+^ channels on the plasma membrane, which are typically blocked by Cs^+^, TMEM175 channels are more permeable to Cs^+^ than to K^+^. Second, the ion conductance of TMEM175 channels is unaffected by various frequently used K^+^ channel blockers, including Ba^2+^, tetraethylammonium, and quinine. Third, in contrast to canonical K^+^ channels, Zn^2+^ modestly inhibits bacterial TMEM175 channels and blocks the human TMEM175 channel with an IC50 of ~38 μM [23,29].

4-aminopyridine (4-AP), a commonly used K^+^ channel blocker, is the only known small-molecule inhibitor that inhibits the human TMEM175 activity, whereas the bacterial TMEM175 channels are insensitive to 4-AP [23]. 4-AP inhibits the hTMEM175-mediated K^+^ conductance with an estimated IC50 of 35 μM [23] and the proton flux through hTMEM175 with an IC50 of 55 ± 13 μM [62]. 4-AP binds in the ion permeation channel close to the isoleucine constriction, precluding the flow of ions and water, as shown by the structure of hTMEM175 bound to 4-AP [62]. The proton current of hTMEM175 was also significantly inhibited by Zn^2+^ [32]. Therefore, both 4-AP and Zn^2+^ were found to inhibit the K^+^ and H^+^ conductance of hTMEM175, supporting the idea that K^+^ and H^+^ share a common ion permeation pathway.

## 4. Physiological Functions of TMEM175

### 4.1. TMEM175 Controls Lysosomal Membrane Potential

An electrical potential gradient (ΔΨ = V_cytosol_ − V_lumen_) exists across the lysosomal membrane due to a stark difference in ion strength between cytosol and lysosomal lumen. Lysosomal ion channels play an essential role in controlling lysosomal membrane potential as they mediate the influx/efflux of various ions across the lysosomal membrane, which alters the ionic homeostasis of the lysosomal lumen [3,11,63]. TMEM175 controls lysosomal ΔΨ through its K^+^ conductance. Lysosomes from the TMEM175 knock-out RAW264.7 cells displayed a depolarized ΔΨ relative to the lysosomes from wild-type cells using a whole-organelle patch-clamp. When measured in the lysosomes from wild-type cells, the lysosomal membrane potential was directly correlated with K^+^ concentration, which indicates the ΔΨ’s sensitivity to K^+^. The lysosomes from the TMEM175 knock-out cells were devoid of K^+^ sensitivity [23]. Even though the membrane potential ΔΨ is widely acknowledged as a critical biophysical property of lysosomes, its cellular roles are not well understood. Presumably, the lysosomal membrane potential ΔΨ may regulate luminal pH and catabolite export [11,23].

### 4.2. TMEM175 Contributes to Lysosomal pH Stability

Lysosomes have an acidic lumen, and the luminal pH value is maintained within a narrow range (4.5–5.5). The delicate regulation of lysosomal pH homeostasis requires the functional orchestration of many ion channels and transporters [54,64]. TMEM175 was found to be critical for lysosomal pH stability. Under fed conditions, the lysosomal pH of TMEM175 knock-out RAW264.7 murine macrophages was comparable to that of wild-type cells. After starvation in Earle’s Balanced Salt Solution (EBSS) for two hours, lysosomes from the TMEM175 knock-out cells showed a significant increase in pH value, whereas wild-type cells maintained a normal lysosomal pH [23]. Similar observations were reported in the SH-SY5Y neuroblastoma cell line. The wild-type SH-SY5Y cells maintained a lysosomal pH in both fed and starvation conditions, while knocking out TMEM175 results in the alkalinization of lysosomes under nutrient-depleted conditions [40,65]. Additionally, lysosomal acidification is disrupted by TMEM175 loss in neurons subjected to oxygen–glucose deprivation/reoxygenation, leading to a marked rise in lysosomal pH. Lysosome alkalinization can be reversed via overexpressing TMEM175 [66]. These pieces of evidence unambiguously point to TMEM175’s involvement in the control of lysosomal pH stability.

It is hypothesized that TMEM175 modulates lysosomal pH stability through counterion pathways as it mediates the exit of potassium ions from the lysosomal lumen [23]. Recent studies have shown that TMEM175 functions as a proton leak channel on the lysosomal membrane that modulates the lysosomal pH homeostasis [34]. The steady-state lysosomal pH is determined by the equilibrium between the proton influx driven by V-ATPases and the proton outflow mediated by TMEM175. As more protons exit lysosomes, TMEM175 overexpression induces an alkaline shift in lysosomal pH, whereas TMEM175 deficiency results in hyper-acidification of lysosomes [34]. The role of TMEM175 in H^+^-leak appears to be inconsistent with the finding of hypo-acidification of lysosomes in TMEM175 knock-out cells during starvation and requires further clarification [23,28].

### 4.3. TMEM175 Regulates Autophagosome-Lysosome Fusion

Macroautophagy (autophagy) is a conserved cellular pathway that eukaryotic cells utilize to degrade and recycle misfolded proteins, protein aggregates, and damaged organelles. Autophagic cargoes are sequestered in double-membrane vesicles which are called autophagosomes. The subsequent fusion of autophagosomes and lysosomes results in the formation of autophagolysosomes, and their contents are digested by lysosomal acidic hydrolases [67,68]. Several potassium channels are proposed to play an important part in the regulation of autophagy, including KCNH7 (potassium voltage-gated channel subfamily H member 7) [69] and a mitochondrial K^+^ ATP channel [70].

It was reported that TMEM175 regulates autophagosome–lysosome fusion [23]. To monitor the number of autophagosomes before and after fusion with lysosomes, the autophagosome marker LC3 (microtube-associated protein light chain 3) was tagged with RFP (red fluorescent protein) and GFP (green fluorescent protein). In this assay, autophagosomes before fusion with lysosomes are marked by RFP and GFP signals, whereas autophagosomes after fusion with lysosomes are only indicated by the RFP signals because of the quenching of GFP signals in the lysosomal acidic milieu [71]. The RFP–GFP–LC3 construct was transiently transfected into RAW264.7 macrophages and the starvation condition was used to induce autophagy [23]. The total number of autophagosomes (indicated by RFP puncta) before and after autophagosome–lysosome fusion is similar in wild-type and TMEM175 knock-out cells, indicating that the formation of autophagosomes is not affected by the deficiency of TMEM175. However, TMEM175 knock-out cells showed a significantly decreased GFP/RFP ratio compared with wild-type cells, which suggests that a larger portion of autophagosomes were fused with lysosomes. Similar findings in mouse hippocampal neurons [28] and SH-SY5Y cells support the hypothesis that TMEM175 deficiency speeds up the fusion of autophagosomes with lysosomes. Furthermore, depletion of TMEM175 leads to stagnant clearance of autophagosomes because of impaired lysosomal degradation [40].

### 4.4. TMEM175 Modulates Lysosomal and Mitochondrial Function

As a lysosomal potassium/proton channel, TMEM175 regulates lysosome physiology in many facets, such as membrane potential and pH homeostasis. Perturbations in lysosomal pH markedly change the catalytic activity of lysosomal hydrolases. The enzymatic activity of cathepsin B, cathepsin D, and glucocerebrosidase significantly decreased in TMEM175 knock-out cells relative to wild-type cells [34,40]. Consistent with this observation, a loss-of-function mutant of TMEM175, p.M393T, was associated with decreased glucocerebrosidase activity [72]. Therefore, TMEM175 deficiency may result in impaired lysosomal degradation. Both cathepsin B and cathepsin D are essential in the lysosomal degradation of α-synuclein [73,74] and decreased glucocerebrosidase is related to the abnormal accumulation of α-synuclein in Parkinson’s disease [75]. Using a preformed α-synuclein fibril (PFF) model, several studies have demonstrated that TMEM175 deficiency causes α-synuclein accumulation in neurons [28,34,40,65]. The TMEM175 knock-out neurons showed much greater aggregation of phosphorylated α-synuclein compared with wild-type neurons, whereas the intensity of phosphorylated α-synuclein inclusions decreased in TMEM175-overexpressed cells, suggesting that TMEM175 has a potential to enhance lysosomal degradation of α-synuclein aggregates.

Besides, TMEM175 may also regulate mitochondrial function. Compared with wild-type rat hippocampal neurons, TMEM175-deficient neurons showed compromised mitochondrial respiration capacity, decreased ATP levels, and reduced migratory ability of active mitochondria [40]. Another study found that TMEM175 overexpression significantly improved mitochondrial activity and ameliorated mitochondrial dysfunction in the course of ischemic injury. Upregulation of the TMEM175 level dramatically reduced the production of reactive oxygen species (ROS), an indicator of mitochondrial dysfunction [66]. A different study, however, presented conflicting results. Overexpression of TMEM175 increased ROS levels and resulted in mitochondrial dysfunction [33]. TMEM175 is primarily localized to lysosomes and endosomes and does not show mitochondrial localization [40]. Accordingly, it is proposed that TMEM175 regulates mitochondrial function through lysosome-mediated mitophagy, the selective degradation of mitochondria via autophagy [76], which plays a key role in the clearance of damaged mitochondria [33,40].

## 5. Regulation of the TMEM175 Channel Activity

Numerous factors, such as ions and membrane potential across the lysosomal membrane, endogenous lipids, luminal acidic pH, and ion channel binding partners, can regulate the activity of lysosomal ion channels [3,11]. The proton conductance of TMEM175 is activated by the luminal acidic pH, which drives the efflux of H^+^ across lysosomal membranes and thereby avoids the over-acidification of lysosomes. An endogenous lipid (arachidonic acid) and two other synthetic chemicals (DCPIB and ML 67–33) were also shown to activate the TMEM175 channel, causing an alkaline shift in the set-point pH of lysosomes [34].

TMEM175 is a growth-factor-activated K^+^ channel and is gated by AKT [28]. AKT binds to the TM2-TM3 linker region of repeat II, which is close to the cytosolic end of the pore-forming TM1 helix and forms a complex with TMEM175 (Figure 3A). AKT regulates TMEM175 channel activity in a kinase-independent manner. Instead, conformational changes in TMEM175-bound AKT are sufficient to activate TMEM175. AKT has a pleckstrin homology (PH) domain at the N-terminus, a kinase domain (KD) in the middle, and a hydrophobic motif at the C-terminus. It is proposed that in the inactive (“PH-in”) conformation, the AKT PH domain negatively regulates AKT kinase activity through intramolecularly interacting with the KD. Upon growth factor activation, the phosphatidylinositol (3,4,5)-trisphosphate [PtdIns(3,4,5)P_3_] level increases on the plasma membrane. PtdIns(3,4,5)P_3_ binding via the AKT PH domain causes conformational changes in AKT, leading to its active (“PH-out”) conformation [77]. A bath application of PtdIns(3,4,5)P_3_ was sufficient to activate TMEM175 current. In contrast, allosteric inhibitors such as MK-2206 and AKT inhibitor VIII that inhibit this conformational change blocked TMEM175 activity [28].

According to a recent study, Bcl-2, an antiapoptotic protein, interacts with TMEM175 and inhibits TMEM175 activity (Figure 3A) [33]. Bcl-2-specific inhibitors increase TMEM175-mediated K^+^ currents in a caspase-independent way. Overexpression of Bcl-2 in HEK293T cells significantly decreased whole-cell TMEM175 currents. Two amino acid residues of TMEM175, R377 and V145, are important to the association between TMEM175 and Bcl-2. Mutations of these two residues abolished their association and relieved the inhibitory effect of Bcl-2 on TMEM175 [33]. Bcl-2 is a critical regulator of mitochondrial apoptosis and is mainly localized on mitochondria. The interactions between mitochondrial Bcl-2 and lysosomal TMEM175 provide additional evidence of communication between mitochondria and lysosomes [78,79]. Additionally, beyond its functional roles in the endosomal–lysosomal pathway, TMEM175 may be implicated in the regulation of mitochondrial functions.

Besides AKT and Bcl-2, TMEM175 has other binding partners that may regulate its function (Figure 3A). According to the BioPlex 3.0 Interactome [80], a human interactome derived from HEK293T and HCT116 cells, TMEM175 was found to interact with SAR1B, TMEM154, TNFRSF10C, and TTYH1. Other potential binding partners include APP [81], GPR37 [82], and two viral proteins (E5 and E5A) [83]. A Calmodulin Target Database scan revealed that TMEM175 contains a single calmodulin-binding domain [84], which indicates that TMEM175 may interact with calmodulin. Moreover, the STRING database (https://string-db.org/ (accessed on 23 December 2022)) was used to analyze the protein–protein interaction networks of TMEM175 (Figure 3B) [85]. TMEM175 has connections with DGKQ, FAM47E, GAK, LRRK2, MCCC1, SMPDL3B, SNRPC, TMEM17, and TMEM163. As previously noted, the *TMEM175/GAK/DGKQ* locus has been identified as a risk locus in PD. Notably, TMEM175 may interact with LRRK2 (leucine-rich repeat kinase 2), an extensively studied PD causative protein [86]. However, their interactions need to be experimentally confirmed. Machine learning algorithms were utilized to identify gene–gene interactions in the Parkinson’s Progression Markers Initiative dataset. A variant rs3822019 in gene *TMEM175* showed a statistically significant interaction with a variant rs17022452 in gene *GAPDHP25* (glyceraldehyde-3 phosphate dehydrogenase pseudogene 25) [87], but it is unclear if they associate with each other at a protein level.

The human TMEM175 is described as a lysosomal potassium/proton channel but HEK293T cells overexpressing GFP-TMEM175 showed detectable GFP signals on their plasma membranes and functional TMEM175 currents can be measured on the plasma membrane [33]. Additionally, the heterologous expression of mouse TMEM175 in *Xenopus laevis* oocytes results in very small currents through the plasma membrane of oocytes. Dynasore compounds, which are potent dynamin inhibitors and disrupt clathrin-dependent endocytosis, have been reported to control the cellular localization of TMEM175 proteins [88]. TMEM175 currents in the plasma membrane were greatly increased in the cells treated with dynasore, with an estimated EC_50_ of at least 30 μM [88]. Dyngo-4a, a hydroxylated derivative of dynasore, induces more pronounced TMEM175 currents with an estimated EC_50_ of 2.3 μM [88]. Following the removal of dynasore chemicals, TMEM175 currents rapidly decreased, indicating the utilization of an effective internalization mechanism to translocate TMEM175 proteins from the plasma membrane to intracellular organelles.

## 6. The Pathological Relevance of TMEM175

### 6.1. TMEM175 Is Identified as a Risk Gene in Parkinson’s Disease

Expression of TMEM175 has been detected in several human tissues with its top three highest expression levels estimated in the cerebellar hemisphere, cerebellum, and pituitary gland [89] (Figure 4). Brain proteome-wide and transcriptome-wide association studies found that TMEM175 shows enrichment in glutamatergic neurons [90]. The high expression level in brains and enrichment in neurons may indicate that TMEM175 is crucial for the neural system’s functions.

Genome-wide association studies (GWASs) investigate the genome of a large group of people and search for single nucleotide polymorphisms (SNPs) that are associated with a particular disease. A GWAS of Parkinson’s disease has identified the *TMEM175/GAK/DGKQ* region at chromosome 4p16.3 as a significant risk locus of this disease [35,36,91,92]. GAK (cyclin G-associated kinase) is a serine/threonine kinase involved in clathrin-mediated membrane trafficking and maintenance of proper centrosome maturation [93,94]. Additionally, GAK was identified as a binding partner of LRRK2 [95]. GAK and a few other interactors of LRRK2 form a complex with LRRK2 promoting removal of *trans*-Golgi-derived vesicles through an autophagy-dependent mechanism. The *DGKQ* gene encodes diacylglycerol kinase theta, which converts diacylglycerol into phosphatidic acid and regulates the respective levels of these two lipids [96]. Using short hairpin RNAs (shRNA) to knock down each of the genes in this locus, *TMEM175* was found to be the only gene that is consistently associated with changes in levels of phosphorylated α-synuclein.

TMEM175 may have a neuroprotective effect, as homozygous TMEM175 knock-out mice at 18–22 months of age showed a significant loss of dopaminergic neurons in the *substantia nigra pars compacta*, which is a pathological feature of PD [28]. In this study, mouse brains from naturally aging mice without any drug treatment were used for neuron counting analysis. However, a recent study presented contradictory results, showing that more dopaminergic neurons were found in TMEM175 knock-out mice compared with wild-type mice when challenged with the neurotoxin MPTP, which suggests that the knock-out of TMEM175 has a neuroprotective effect [33]. Conflicting results reported in these studies may be attributed to the use of different mouse models. The naturally aging mouse model mimics the Parkinsonism caused by genetic factors, whereas the MPTP-induced mouse model mimics the Parkinsonism caused by environmental factors. Further studies are needed to investigate the pathological roles of TMEM175 in different types of PD.

### 6.2. Two Important SNPs of TMEM175 in Parkinson’s Disease

Further investigation of the chromosome 4: 951,947 GWAS signal in this PD-related region reveals a missense SNP in TMEM175, rs34311866 (p.M393T), which is the most significant coding variant in this locus. Conditional analysis of this main SNP results in the identification of a secondary genome-wide significant SNP, rs34884217 (p.Q65P) (Table 2) [65,72,97].

The p.M393T and p.Q65P variants are independent as they are not in linkage disequilibrium [72]. TMEM175 (M393T) is a loss-of-function mutant. Compared with the wild-type TMEM175, the M393T mutation decreases lysosomal K^+^ currents by about 40% in HEK293T cells [28]. In contrast, TMEM175 (Q65P) is a gain-of-function mutant under stress. One hour of starvation is sufficient to abolish the K^+^ current in wild-type TMEM175- and TMEM175 (M393T)-expressing HEK293T cells, whereas it has little effect on the K^+^ current of TMEM175 (Q65P)-expressing HEK293T cells. Additionally, these two mutations also have different effects on the proton conductance of TMEM175. The Q65P mutation promotes the proton conductance of TMEM175 while the M393T mutation decreases it [32]. As the proton conductivity of TMEM175 is critical to maintaining lysosomal pH stability and lysosomal hydrolases require a narrow pH for their optimal activity, it may explain why the M393T mutant shows decreased glucocerebrosidase activity relative to wild-type TMEM175 [65,72], whereas the Q65P mutation does not affect the glucocerebrosidase activity [72].

According to a meta-analysis of variants in three PD cohorts, the M393T mutation in TMEM175 is associated with an increased risk for PD, while the Q65P mutation is associated with a decreased risk for PD [72]. It has been reported that the TMEM175 M393T mutation affects the progression of PD. Through studying a cohort of 341 longitudinally followed PD patients, the M393T mutation was found to be associated with the rate of cognitive decline [28] and the rate of motor decline in PD, with p.M393T carriers declining more rapidly than patients without this mutation [28,98]. Moreover, multiple GWASs and meta-analysis of age at onset of PD have identified the TMEM175 rs34311866 (p.M393T) as a genome-wide significant variant that is associated with earlier age at onset of PD [72,92,99,100]. In a cohort of 1526 Danish PD patients, the *TMEM175* rs34311866 variant showed a decrease in onset age by 1.2 years (95% confidence interval, 2.0–0.4) per PD risk allele [101].

Structural analysis of TMEM175 M393T and Q65P mutants revealed that these two mutations may have an impact on the structure of TMEM175. The C-terminal transmembrane domain of TMEM175 is proposed to be significantly destabilized by the M393T mutation, whereas the Q65P mutation only induces a minor global destabilization [72]. More structural and functional studies are needed to fully decipher how these two mutations influence the functions of TMEM175.

### 6.3. SNPs of TMEM175 in Other Diseases

The TMEM175 p.M393T mutation, in addition to being associated with a higher risk for PD, is also an independent risk locus for rapid eye movement sleep behavior disorder, a prodromal syndrome of alpha-synucleinopathies (Table 2) [72,102]. Other SNPs of TMEM175 were shown to be associated with early-onset and late-onset PD [103,104]. Genome-wide association studies and mutation analysis have identified important SNPs of *TMEM175* in many other diseases such as type 2 diabetes [105], breast cancer [106], systemic sclerosis [107], Lewy body dementia [108], and short QT syndrome [109] (Table 2). However, the functional role of TMEM175 in these diseases remains unclear. Remarkably, *TMEM175* is a critical genetic risk locus for Parkinson’s disease, Lewy body dementia, and rapid eye movement sleep behavior disorder, showcasing its significance in maintaining the proper functions of neurons.

**Table 2 biomolecules-13-00802-t002:** SNPs of *TMEM175* in diseases.

SNP/Mutation	Genomic Position (bp)	Reference Allele	Alternate Allele	Disease	References
rs34311866	Chr.4:951947	T	C	PD	[35]
rs34311866	Chr.4:951947	T	C	RBD	[72,102]
rs34884217	Chr.4:950422	A	C	PD	[35]
rs2290402	Chr.4:931518	T	C	Type 2 diabetes	[105]
rs2290405	Chr.4:953186	G	A, C	Breast cancer	[106]
rs2290405	Chr.4:953186	G	A, C	Systemic sclerosis	[107]
rs6599388	Chr.4:945299	C	T	Lewy body dementia	[108]
c.A526G	Chr.4:949608	A	G	Early-onset PD	[103]
c.C547T	Chr.4:947062	C	T	Late-onset PD	[104]
p.Arg377His	-	-	-	Short QT Syndrome	[109]

Abbreviations: PD, Parkinson’s disease; RBD, rapid eye movement sleep behavior disorder.

## 7. Conclusions and Perspectives

TMEM175 is a pore-forming protein that was initially characterized as a K^+^-selective channel in endosomal–lysosomal membranes. Its channel activity is essential for the regulation of lysosomal membrane potential, lysosomal pH stability, and lysosome–autophagosome fusion. Dysfunction or loss of TMEM175 leads to impaired lysosomal degradation, resulting in compromised hydrolase activity, accelerated lysosome–autophagosome fusion, and accumulated pathogenic α-synuclein. Recent studies have shown that TMEM175 also mediates proton conductance and functions as a lysosomal H^+^-leak channel, modulating the acidity of lysosomes.

Human and bacterial TMEM175 channels show distinct cellular localization. When expressed in HEK293 cells, the bacterial MtTMEM175 enters the secretory pathway and is finally sorted into the plasma membrane [30]. Overexpression of human TMEM175 in HEK293T cells [33] or mouse TMEM175 in *Xenopus laevis* oocytes [88] generates detectable currents on the plasma membrane. Moreover, dynasore, an endocytosis inhibitor, is able to translocate lysosomal TMEM175 channels to the plasma membrane [88]. It is therefore tempting to hypothesize that TMEM175 is also found on the plasma membrane of native cells and goes through an internalization process before arriving at its target organelles, endosomes and lysosomes. Further research should be undertaken to investigate why TMEM175 is translocated to the plasma membrane, how TMEM175 gets sorted into endosomes and lysosomes, and whether it performs any biological functions there.

The pathological roles of TMEM175 in PD are primarily underpinned by its modulation of the endo-lysosomal pathway because it is a lysosomal potassium and proton channel that regulates the physiology of lysosomes. TMEM175 is also shown to be involved in the apoptotic pathway. The apoptosis regulator Bcl-2 inhibits the TMEM175 channel activity through direct binding. TMEM175 is a proapoptotic ion channel that plays an important role in the death of neurons [33]. Protein–protein interaction analysis revealed that TMEM175 has other potential binding partners. Therefore, it is important to find out whether they are TMEM175 modulators as this could shed light on TMEM175’s roles in other cellular pathways.

Potassium channels play a significant role in PD pathophysiology and represent an appealing therapeutic target for PD [110,111]. The *TMEM175* gene has been identified as a risk locus of PD. TMEM175 or its knock-out exhibit neuroprotective effect in different PD mouse models. Finding specific activators or blockers may be a prospective strategy for PD treatment. It’s important to note that TMEM175 has been linked to a number of different neurological conditions, including Lewy body dementia and rapid eye movement sleep behavior disorder. Additionally, many SNPs of TMEM175 are shown to be associated with other diseases, such as breast cancer and systemic sclerosis. Further work is required to confirm the involvement of TMEM175 variants in these diseases and define their pathological roles.

## Figures and Tables

**Figure 1 biomolecules-13-00802-f001:**
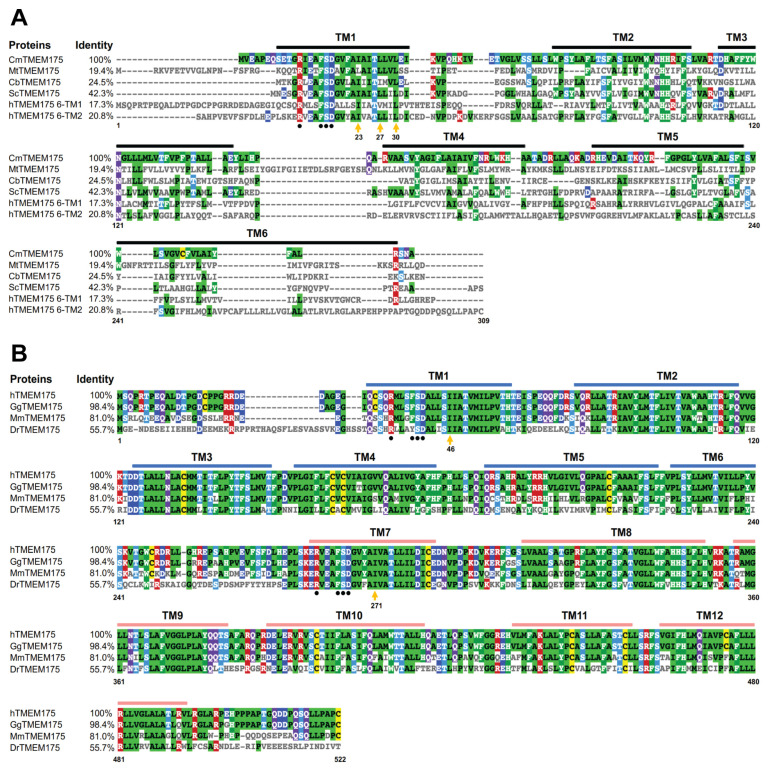
Sequence alignment analysis of prokaryotic and eukaryotic TMEM175 proteins. (**A**) Sequence alignment analysis of prokaryotic TMEM175 proteins and the human TMEM175 repeat I and repeat II. (**B**) Sequence alignment analysis of eukaryotic TMEM175 proteins. The letters before TMEM175 indicate the species where it comes from, and UniProtKB reference numbers for these TMEM175 proteins are shown. Cm, *Chamaesiphon minutus*, K9UJK2; Mt, *Marivirga tractuosa*, E4TN31; Cb, *Chryseobacterium* sp., A0A086F3E3; Sc, *Streptomyces collinus*, S5VBU1; h: *Homo sapiens*, Q9BSA9; Gg, *Gorilla gorilla*, G3R453; Mm, *Mus musculus*, Q9CXY1; Dr, *Danio rerio*, A5PN43. The Clustal Omega sequence alignment program was used to align amino acid sequences and the sequence alignment result was shown and colored using MView.

**Figure 2 biomolecules-13-00802-f002:**
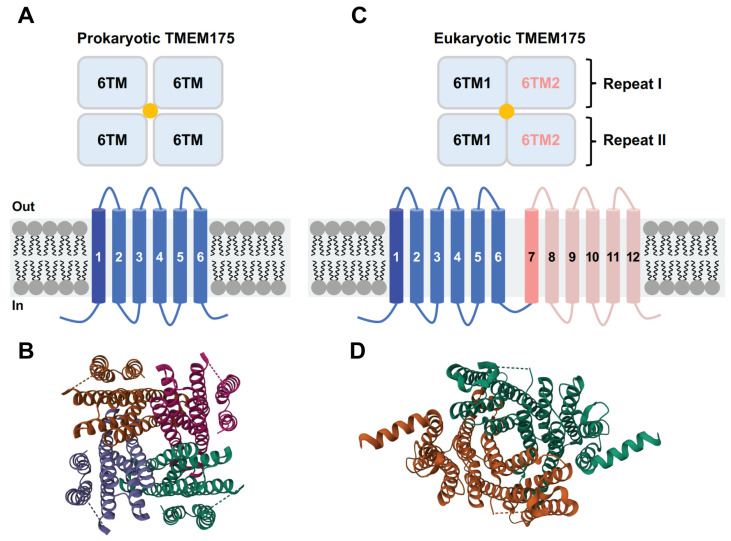
Topological arrangement of prokaryotic and eukaryotic TMEM175 channels. (**A**) Schematic diagram of the prokaryotic TMEM175 channel structures. (**B**) Structural model of CmTMEM175 (PDB code: 5VRE). (**C**) Schematic diagram of the eukaryotic TMEM175 channel structures. (**D**) Structural model of hTMEM175 (PDB code: 6WC9, open state TMEM175 in KCl).

**Figure 3 biomolecules-13-00802-f003:**
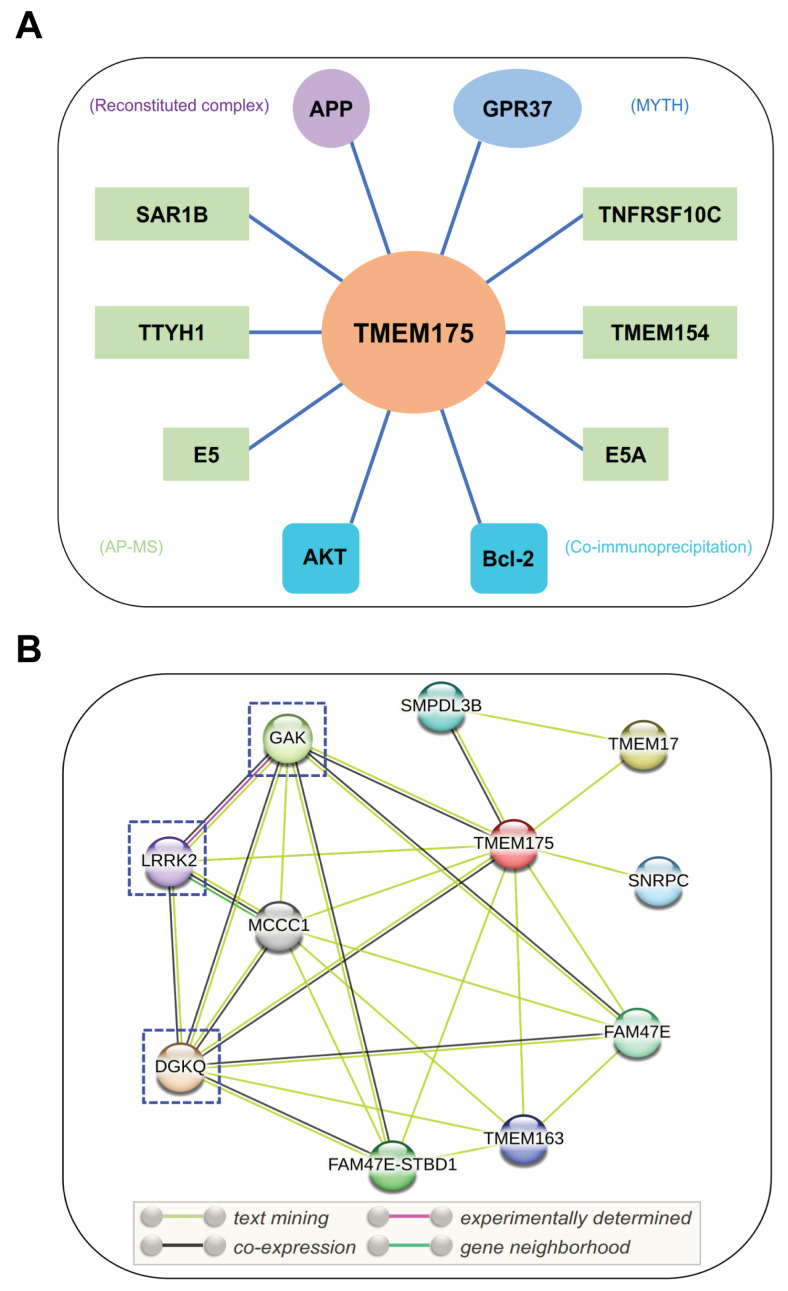
Interacting partners of TMEM175. (**A**) Potential binding partners of TMEM175. The experimental methods used to identify these TMEM175-protein interactions are shown in brackets. GPR37, G-protein-coupled receptor 37; APP, amyloid-beta precursor protein; E5 and E5A, viral proteins of human papillomavirus; SAR1B, secretion-associated Ras-related GTPase 1B; TMEM154, transmembrane protein 154; TNFRSF10C, tumor necrosis factor receptor superfamily member 10c; TTYH1, Tweety Homolog 1. MYTH, modified membrane yeast two-hybrid; AP-MS, affinity purification mass spectrometry. (**B**) STRING analysis of TMEM175-protein interaction networks. DGKQ, diacylglycerol kinase theta; FAM47E, family with sequence similarity 47 member E; GAK, cyclin G-associated kinase; LRRK2, leucine-rich repeat kinase 2; MCCC1, methylcrotonoyl-CoA carboxylase subunit alpha; SMPDL3B, acid sphingomyelinase-like phosphodiesterase 3b; SNRPC, U1 small nuclear ribonucleoprotein C; FAM47E-STBD1, FAM47E-STBD1 readthrough; TMEM17, transmembrane protein 17; and TMEM163, transmembrane protein 163. The PD-related binding partners of TMEM175 are highlighted by dashed boxes.

**Figure 4 biomolecules-13-00802-f004:**
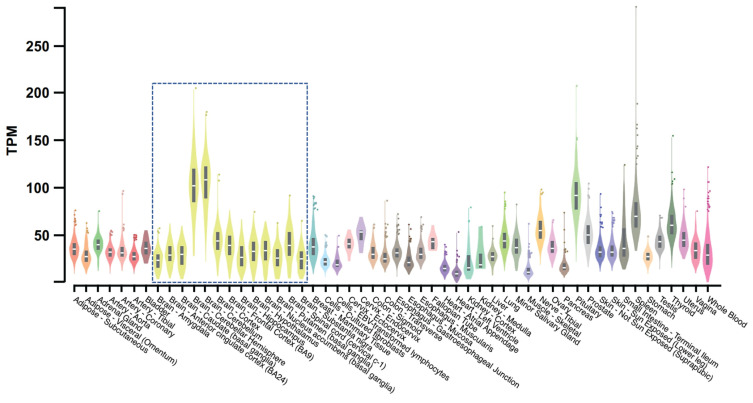
Bulk tissue gene expression for TMEM175. This figure is processed from the GTEx portal (https://www.gtexportal.org (accessed on 1 December 2022)) [89]. Data source: GTEx Analysis Release V8 (dbGaP Accession phs000424.v8.p2). The gene expression level is shown in TPM (transcripts per million kilobases), calculated from a gene model with isoforms collapsed to a single gene. The gene expression level in different tissues is indicated by different colors with yellow violin plots (boxed) showing expression levels in brain tissues.

## Data Availability

The data presented in Figure 1 are openly available in GTEx Portal (https://www.gtexportal.org/home/ (accessed on 1 December 2022)). For the STRING protein–protein interaction network analysis, the data are available in the STRING database (https://string-db.org/ (accessed on 23 December 2022)).

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
