# Peer review of "Transmembrane Protein 175, a Lysosomal Ion Channel Related to Parkinson’s Disease"

_biomolecules, 2023, doi:10.3390/biom13050802_

Round 1
Reviewer 1 Report
The review article by Tang and colleagues is a well-written literature overview on the molecular aspects of the lysosomal ion channel TMEM175. My overall view is that the structural, genetic and molecular aspects are covered really well and in much detail. However, the author title the article with its relation with Parkinson's disease, which is present in a very limited manner in the actual text. My suggestion is to follow one of two routes:
- move the focus on TMEM175 itself and highlight the physiological function, mentioning in general its relations to diseases;
- expand the section on Parkinson's disease, including hypotheses from the authors on its involvement in etiology and possibilities for development of therapeutic targeting. This latter part is completely lacking in the manuscript.
Below some other specific points which I believe require attention:
- Lines 239-243, the interpretation of the GFP-RFP-LC3 tandem construct experiments are a bit confusing. Autophagosomes are labeled by colocalized GFP and RFP, while RFP alone indicates autolysosomes. A reduction in GFP puncta (considered alone) indicates more autolysosomes but does not give information on the amount of autophagosomes to begin with. I suggest the author to inspect deeper the referenced article;
- Lines 299-301, GAK is reported to be involved in LRRK2-dependent modulation of Trans-Golgi Network trafficking, possibly involving lysosomes (Beilina et al., 2014). This should be mentioned and discussed, and not quickly dismissed;
- Lines 275-278, and 496-500, are there any studies specifically looking at the role of TMEM175 in mitophagy? It would be nice to have the authors' hypothesis/speculation on this, especially in the context of diseases.
Reviewer 2 Report
Tang and colleagues summarized the structure, physiological functionality, regulation, and PD relevance of transmembrane protein 175 (TMEM175). The authors introduced the TMEM175 in the context of a lysosomal potassium channel and emphasized its uniqueness in genetics, ion selectivity, and control of lysosomal pH stability. The authors detailed two relevant SNPs of TMEM175 in PD and listed additional significant SNPs in other diseases. This review is comprehensive and up-to-date, yet its structure is inorganic as a patchwork of independent sections.
Specific Points:
1. Sections were not in order. The introduction mapped out the structure of this review as the lysosome function, lysosomal ion channels, lysosomal K+ ion channels, TMEM175, and TMEM175 in PD, in the order of specificity. However, the paper did not follow this map but jumped from the pathological relevance of TMEM175 (Section 3) to prokaryotic vs. eukaryotic structure (Section 4).
2. Section 5 summarised the K+ selectivity and inhibition pharmacology. This section should be part of the Ion selectivity of TMEM175 in section 1. In addition, the proposed common ion (K+ and H+) permeation pathway in section 5 fits the idea in 1.2 that hTMEM175 is a lysosomal H+-leak channel.
3. Section 6 introduced the regulation of TMEM175 (AKT, Bcl-2, binding partner analysis). This section should be part of the physiological function of TMEM175 in section 2 since the regulation of TMEM175-mediated K+-current implicated its physiological role in regulating lysosomal and mitochondrial functions.
4. The authors should elaborate on the competing views on TMEM175's contribution to lysosomal pH stability. In sections 1.2 and 2.2, the authors list evidence supporting that hTMEM175 is a lysosomal H+-leak channel. Whereas the results from TMEM175 KO cells in references 23, 28, 40, 57, and 58 support an inconsistent role of TMEM175 in lysosomal stability. The authors need to critically present both views for the readers to appreciate emerging discoveries on this topic.
5. Figure 4B SRING analysis of TMEM175 is reader unfriendly. The color scheme is hard to distinguish different types of interactions. The authors also need to emphasize the PD-related connections in Figure 4.
6. The authors overwhelmingly rely on reference 23 for TMEM175's measurements and functionality and cite over 19 times throughout the paper.
7. Lines 284-286, the authors stated that TMEM175 mRNA 'increased' expression levels in mouse astrocytes (Reference 71). The authors should clarify which cell type TMEM175 mRNA level in astrocytes was compared with.
8. The authors incorrectly list rapid eye movement sleep behavior disorder as one of the alpha-synucleinopathies. While the REM sleep behavior disorder may be associated with Lewy body dementia, PD, or MSA, it is not a kind of alpha-synucleinopathies.
9. The authors need to strengthen the link between TMEM175 and PD. They need to expand section 2.4 on TMEM175's role in alpha-synuclein degradation (reference 28, 34, 40, and 57), directly related to PD development. They also need to elaborate on the conflict results from reference 33, in which the TMEM175 overexpression aggravates symptoms of PD in the model.
Reviewer 3 Report
Here the authors have written an extensive review article about the unique lysosomal K+ channel, TMEM175. TMEM175 shares little sequence similarity to other potassium channels and is one of three lysosomal K+ channels. The conductance mediated by TMEM175 is important for setting membrane potential, maintaining pH (critical for functional lysosomes) and regulating lysosome-autophagosome fusion. GWAS studies have established that TMEM175 is implicated in the pathogenesis of Parkinson’s disease and the impact of TMEM175 KO and missense mutations on alpha-synuclein phosphorylation and aggregate accumulation has been studied. Overall, I think the authors have summarized the topic very well and highlighted the importance of the phyioslogical relevance of TMEM175 channels but also their link to Parkinson’s disease. I have a few comments and questions listed below:
1) Personally, I think the paragraph describing the structure of the channels would be more beneficial early in the review (perhaps right after the introduction) as it helps the reader visualize whilst then reading onwards.
2) I felt that the paragraph on the pharmacology of TMEM175 was a bit weak and perhaps not the most relevant to the topic of TMEM175 involvement in PD. The reason being that it is from an inhibitor point-of-view and it seems that inhibiting it (with relation to PD) is not what one would want. Could the authors give a perspective on how it could be targeted more so for PD, insights into activation perhaps? How you would balance this without disrupting normal physiological functions? Etc.
3) In the conclusion and perspective, the authors raise some interesting work on the affects to dopaminergic neurons (that have some contradictory findings) – I think this should be discussed more fully earlier in the review (the section about TMEM175 in PD). If they can also comment on the extenuating circumstances in the studies – the one that showed a positive effect was in the present of a neurotoxin; it’s noted that with the Q65P mutation, starvation conditions do not affect K+ current where it does for WT and KO. Some context about the study conditions would be helpful in understanding the complexities.
4) In the final conclusions, can the authors perhaps give a more global discussion about targeting K+ ion channels (more generally) for Parkinson’s disease?
Round 2
Reviewer 2 Report
The authors reorganized the sections of the paper, making it more readable. They also addressed all the concerns in the last review and strengthened the link between TMEM175 and Parkinson's disease in conclusion. The quality of the manuscript has been significantly improved.